# A Novel Non-Specific Lipid Transfer Protein Gene, *CmnsLTP6.9*, Enhanced Osmotic and Drought Tolerance by Regulating ROS Scavenging and Remodeling Lipid Profiles in Chinese Chestnut (*Castanea mollissima* Blume)

**DOI:** 10.3390/plants12223916

**Published:** 2023-11-20

**Authors:** Yuxiong Xiao, Cui Xiao, Xiujuan He, Xin Yang, Zhu Tong, Zeqiong Wang, Zhonghai Sun, Wenming Qiu

**Affiliations:** Hubei Key Laboratory of Germplasm Innovation and Utilization of Fruit Trees, Institute of Fruit and Tea, Hubei Academy of Agricultural Sciences, Wuhan 430064, China; yuxiongx@163.com (Y.X.); xiaocui@hbaas.com (C.X.); hexiujuan@hbaas.com (X.H.); 15310308451@163.com (X.Y.); tongzhu@hbaas.com (Z.T.); wangzq@hbaas.com (Z.W.); hbfruit@126.com (Z.S.)

**Keywords:** chestnut, *CmnsLTP*, *CmnsLTP6.9*, osmotic, drought

## Abstract

Chestnut (*Castanea mollissima* Blume) is an important economic tree owing to its tasty fruit and adaptability to environmental stresses, especially drought. Currently, there is limited information about non-specific lipid transfer protein (nsLTP) genes that respond to abiotic stress in chestnuts. Here, a chestnut *nsLTP*, named *CmnsLTP6.9*, was identified and analyzed. The results showed that the CmnsLTP6.9 protein localized in the extracellular matrix had two splicing variants (*CmnsLTP6.9L* and *CmnsLTP6.9S*). Compared with *CmnsLTP6.9L*, *CmnsLTP6.9S* had an 87 bp deletion in the 5′-terminal. Overexpression of *CmnsLTP6.9L* in *Arabidopsis* enhanced tolerance to osmotic and drought stress. Upon exposure to osmotic and drought treatment, *CmnsLTP6.9L* could increase reactive oxygen species (ROS)-scavenging enzyme activity, alleviating ROS damage. However, *CmnsLTP6.9S*-overexpressing lines showed no significant differences in phenotype, ROS content, and related enzyme activities compared with the wild type (WT) under osmotic and drought treatment. Moreover, lipid metabolism analysis confirmed that, unlike *CmnsLTP6.9S*, *CmnsLTP6.9L* mainly altered and upregulated many fatty acyls and glycerophospholipids, which implied that *CmnsLTP6.9L* and *CmnsLTP6.9S* played different roles in lipid transference in the chestnut. Taken together, we analyzed the functions of *CmnsLTP6.9L* and *CmnsLTP6.9S*, and demonstrated that *CmnsLTP6.9L* enhanced drought and osmotic stress tolerance through ROS scavenging and lipid metabolism.

## 1. Introduction

Chestnut (*Castanea mollissima* Blume), native to China, belongs to the Chestnut genus in the Fagaceae family. Nowadays, it is widely distributed in China due to its important economic and ecological value. Chestnuts are rich in nutrition such as starches, sugars, proteins, and fats. In addition to the kernels used for food processing, chestnut shells, flowers, and leaves have been widely used for medical, biomass, and catalyst materials [1,2,3]. As chestnut trees are resistant to barrenness and conserve water, they have been extensively used for the afforestation and greening of barren mountains. Environmental stresses such as salinity, osmosis, drought, heavy metals, low temperatures, and disease lead to a decline in the quality and yield of agricultural products. Drought stress decreases photosynthesis, yield, and carbon/nitrogen metabolism in Chinese chestnut ‘Yanshanzaofeng’ seedlings [4]. Meanwhile, water stress affects the cell membrane lipid oxidation and calcification of chestnut [5]. At present, reports on chestnuts mainly focus on storage, processing, and pest/disease control; research on abiotic stress is relatively scarce [6,7,8]. As most chestnuts grow in barren areas, drought and osmotic stress are two common abiotic stresses that limit the growth and development of chestnuts and affect fruit quality and yield. However, there are relatively few studies on the physiological and molecular mechanisms of drought and osmotic stress in chestnuts.

It is well known that in response to drought and osmotic stress, the morpho-physiological and tolerance mechanisms of plants include the regulation of ionic and osmotic homeostasis, reactive oxygen species (ROS) detoxification, leaf photosynthesis, hormonal and gene expression regulation, and secondary metabolism [9,10,11,12]. Lipids, as an important secondary metabolite, play a crucial role in plants’ response to environmental stress, such as membrane stabilization, plant extracellular vesicle function, and cuticle and cell wall integrity [13,14,15,16]. In plants, considerable evidence shows that nsLTPs are involved in a range of lipid-related biological processes, including oil accumulation, cuticular wax deposition, pollen wall formation, and cutin synthesis [13,17,18,19], while *nsLTPs* have also been found to respond to a variety of abiotic/biotic stresses, such as drought, high salinity, heat, cold stress, pathogen defense, PEG 6000 stress, and IAA and ABA stresses [20,21,22,23,24,25]. For abiotic stress, *OsLTPL159* increased cold tolerance at the early seedling stage in rice [26]. *NtLTPI.38* enhanced salt tolerance in tobacco by regulating lipid and flavonoid synthesis, antioxidant activity, ion homeostasis, and ABA signaling pathways [27]. *GhLTP4* enhanced drought tolerance by remodeling lipid profiles, regulating abscisic acid homeostasis, and improving the tricarboxylic acid cycle in cotton [28]. *TaLTP40* and *TaLTP75* were found to enhance wheat salt tolerance by transferring membrane lipids to biological membranes [29]. In other studies, *ZmLTP3* and *AtLTP3* positively regulated salt and drought stresses, respectively [30,31]. However, the function of nsLTPs in most woody plants remains unclear under abiotic stress. For biotic stress, *NbLTP1* positively regulated plant immunity against viral infection through upregulating SA biosynthesis and suppressing ROS accumulation at the later phases of viral pathogenesis [32]. *StLTP10* positively regulated plant resistance to *Phytophthora infestans* by regulating the expression of ROS scavenging and defense-related genes [33]. Wheat apoplast-localized lipid transfer protein *TaLTP3* enhanced defense responses against *Puccinia triticina* by regulating *TaPR1a* [34]. *AtLTP4.4* was found to be able to combat trichothecene by increasing glutathione-based antioxidant defense [35]. In another study, *Populus trichocarpa* type V LTP (*Potri.008G061800*) functioned as glutathione-S-transferase and could weakly inhibit the growth of *Septotis populiperda* in vitro [36]. Therefore, we hypothesized that plant *nsLTPs* are involved in biotic and abiotic stress mainly through ROS scavenging and lipid transfer.

Considering the importance of investigating the molecular networks, biological processes, and functions of *nsLTPs*, a genome-wide analysis of the *nsLTPs* in chestnut was undertaken. In this study, putative *nsLTPs* were identified, and their phylogenetic trees, gene structures, and expression profiles were determined to evaluate the molecular evolution and biological function of the *nsLTP* family in chestnut. Furthermore, two spliceosomes of *CmnsLTP6.9* (*CmnsLTP6.9L* and *CmnsLTP6.9S*) were cloned from chestnut. At present, the function of *CmnsLTP6.9* has not been reported in plants, including homologous genes in *A. thaliana* and *Brassica napus*. Overexpression of *CmnsLTP6.9L* rather than *CmnsLTP6.9S* in *Arabidopsis* significantly enhanced osmotic and drought tolerance by improving the activities of ROS-scavenging enzymes and regulating lipid metabolism. These results indicated the different functions of *CmnsLTP6.9L* and *CmnsLTP6.9S* in lipid metabolism and osmotic and drought stress. Our study comprehensively analyzes the nsLTP protein family and uncovers potential osmotic and drought-tolerant genes, which would provide valuable candidate genes for the breeding of osmotic and drought-tolerant varieties in chestnut and other plants.

## 2. Results

### 2.1. Phylogenetic Analysis of Chestnut nsLTPs (CmnsLTPs)

A total of 60 *nsLTPs* were identified in chestnut and designated as *CmnsLTPs*. A phylogenetic analysis of these *CmnsLTPs* was conducted using *B. napus* and *A. thaliana nsLTPs* as references. All of the *CmnsLTPs* were divided into nine subfamilies (type I, type II, type III, type IV, type V, type VI, type VIII, type IX, and type XI). Type I was the largest subfamily, containing 13 genes, followed by type V (12 genes), type XI (10 genes), type VI (9 genes), type II (5 genes), type IV (5 genes), type IX (3 genes), and type VIII (2 genes). Type III (1) contained the fewest members (Figure 1). In general, the *CmnsLTPs* of chestnuts are closely related to those of *B. napus*.

### 2.2. Gene Structures and Conserved Domains of CmnsLTPs

We observed that some types of *CmnsLTPs* shared similar conserved motifs (Figure 2). The type I, type VIII, and type IX *CmnsLTPs* shared motif 4 and motif 6; the type II *CmnsLTPs* shared motif 6 and motif 16; the type III, type IV, and type VI *CmnsLTPs* shared motif 6; and the type V *CmnsLTPs* shared motif 4 (Figure 2). Among the motif-containing sequences of 60 *CmnsLTPs*, 46 *CmnsLTPs* contained motif 4, and 38 *CmnsLTPs* contained motif 6, which appeared the most frequently (Figure 2). Thus, it was hypothesized that motif 4 and motif 6 were conserved motifs of the *CmnsLTPs*. In addition, the gene structural diagram of the *CmnsLTPs* was analyzed. All members of types I and XI contained one exon. Members of types I, III, VI, and XI contained one to two exons, except *CmnsLTP1.4* and *CmnsLTP11.4*. Most of the members of types II, V, and IX contained more than three exons (Figure 2). In general, genes from the same type had similar gene structures and conserved motif compositions, which supported the classification of these genes.

### 2.3. Regulatory Elements of CmnsLTPs

In total, 36 cis-acting elements were predicted (Figure 3). These cis-acting elements were mainly related to light, circadian control, cell differentiation, the plant hormones (MeJA, salicylic acid, abscisic acid, auxin, and gibberellin), and biotic and abiotic stresses (wound, defense, low temperature, and drought), suggesting that *CmnsLTPs* may have multiple biological functions (Figure 3).

### 2.4. Expression Profiling of CmnsLTPs

The expression profiles of *CmnsLTPs* in four different tissues, namely the stem, leaf, male flowers, and female flowers were examined. The results indicated that the *CmnsLTPs* were tissue specific, such as *CmnsLTP11.5*, *CmnsLTP11.9*, and *CmnsLTP11.10*, which were highly expressed in the leaf, while *CmnsLTP1.6*, *CmnsLTP1.9*, and *CmnsLTP11.6* were highly expressed in the female flowers. In general, more than half of the *CmnsLTPs* were highly expressed in the male flowers. Among them, *CmnsLTP1.1*, *CmnsLTP1.2*, *CmnsLTP1.10*, *CmnsLTP6.2*, *CmnsLTP6.7*, and *CmnsLTP6.9* were highly expressed during pollen maturation, indicating that these genes may be involved in lipid transport during pollen maturation (Figure 4A).

A total of 11 *CmnsLTPs* specifically expressed in leaves or flowers were screened for further analysis. The results showed that the expression of all 11 *CmnsLTPs* was induced after osmotic and drought stress. After osmotic treatment, more genes were upregulated, except for *CmnsLTP1.6* and *CmnsLTP1.10* (Figure 4B). However, five genes were upregulated and six genes were downregulated during drought treatment (Figure 4B). These results suggest that these *CmnsLTPs* functioned differently upon osmotic and drought stress in chestnut. Excluding downregulated genes, *CmnsLTP1.1*, *CmnsLTP5.1*, *CmnsLTP6.9*, and *CmnsLTP11.5* positively responded to osmosis and drought treatment, in which *CmnsLTP1.1*, *CmnsLTP5.1*, and *CmnsLTP11.5* were highly expressed in the pre- and post-osmosis treatment stages, and only *CmnsLTP6.9* was highly expressed in the early and middle osmosis treatment stages (Figure 4B). Combined with tissue specificity and stress response analysis, it is speculated that *CmnsLTP6.9* played an important role in lipid transport, drought, and osmotic response.

### 2.5. Sequence Characterization and Subcellular Localization of CmnsLTP6.9

The full-length *CmnsLTP6.9* was cloned using the cDNA of male chestnut flowers. The open reading frame (ORF) analysis of *CmnsLTP6.9* showed there were two spliceosomes named *CmnsLTP6.9S* and *CmnsLTP6.9L*, both of them containing a conserved domain AAI_LTSS super family (Figure 5A). *CmnsLTP6.9S* has an 87 bp deletion in the 5′-terminal of *CmnsLTP6.9L*, and the deleted sequence-encoded 29 amino acids (aa) residue contained a putative 24 aa signal peptide (Appendix A). The two spliceosomes were cloned into the binary vector pCAMBIA2300-GFP and fused with a green fluorescent protein (GFP). Transient expressions of *CmnsLTP6.9S*-GFP and *CmnsLTP6.9L*-GFP in onion epidermal cells revealed that the *CmnsLTP6.9S* and *CmnsLTP6.9L* protein localized to the cellular boundary and primarily in the peripheral cell layers (Figure 5B).

### 2.6. Overexpression of CmnsLTP6.9-Enhanced Tolerance to Osmotic and Drought Stress in A. thaliana

Transgenic *Arabidopsis* overexpressing *CmnsLTP6.9S* and *CmnsLTP6.9L* were generated. The transgenic lines were verified by RT-PCR and green fluorescence, of which five lines with the highest expression levels were used for subsequent functional analysis (Appendix A). One-month-old WT and T3 homozygous overexpression (OE) seedlings were grown with exogenous mannitol or without water for another 2 weeks. The WT and OE plants were indistinguishable under the control conditions, but the WT and *CmnsLTP6.9S*-OE plants showed worse symptoms such as leaf wilting and chlorosis in comparison with the *CmnsLTP6.9L*-OE plants under osmotic and drought treatment (Figure 6A). The relative water content was higher, while the relative electrolyte leakage was lower in the *CmnsLTP6.9L*-OE plants than in the WT and *CmnsLTP6.9S*-OE plants (Figure 6B,C). In addition, there was no significant difference in proline content, but the MDA content in the *CmnsLTP6.9L*-OE plants was lower than that in the WT and *CmnsLTP6.9S*-OE plants (Figure 6D,E). Taken together, these results indicate that the overexpression of *CmnsLTP6.9L* improved the tolerance to drought and osmotic stress compared with WT and *CmnsLTP6.9S*-OE.

### 2.7. Overexpression of CmnsLTP6.9 Stimulated ROS Scavenging in A. thaliana under Osmotic and Drought Stress

We examined the accumulation of H_2_O_2_ using 3,3′-diaminobenzidine (DAB) staining and superoxide (O_2_^−^) with nitro blue tetrazolium (NBT) staining. The *CmnsLTP6.9L*-OE transgenic plants displayed a lighter staining pattern for both DAB and NBT compared to WT and *CmnsLTP6.9S*-OE, indicating that the activity of ROS scavenging systems was higher in the *CmnsLTP6.9L*-OE plants (Figure 7A). However, there was no significant difference between the WT and *CmnsLTP6.9S*-OE plants in terms of DAB and NBT staining, implying that *CmnsLTP6.9S* lost the function to reduce ROS (Figure 7A). In addition, the content of H_2_O_2_ was further measured and analyzed, which was consistent with the results of DAB staining. The accumulation of H_2_O_2_ in *CmnsLTP6.9L*-OE was significantly slower than that in the WT and *CmnsLTP6.9S*-OE, and there was no difference between the latter two (Figure 7B). To understand the mechanism of ROS scavenging, we further tested the activities of two antioxidant enzymes, namely SOD and POD. Correspondingly, the SOD and POD activities of the WT and *CmnsLTP6.9S*-OE plants were indistinguishable, and the values for the *CmnsLTP6.9L*-OE plants were significantly higher than the WT and *CmnsLTP6.9S*-OE plants under osmotic and drought stress (Figure 7C,D). These results suggest that the increased resilience of the *CmnsLTP6.9L*-OE plants under osmotic and drought stress was linked to the higher activity of their ROS scavenging systems.

### 2.8. Lipidomic Analysis of Transgenic Arabidopsis

Lipidomic analyses were conducted using *Arabidopsis* transgenic lines (W, wild types; L1, *CmnsLTP6.9L*-OE; L2, *CmnsLTP6.9S*-OE), with the thresholds of adjusted *p* < 0.05, and the differentially expressed lipids (DELs) were identified. L1_vs_L2 (388 DELs) had the most DELs among the three comparisons, with 171 and 217 DELs up- and downregulated, respectively. In contrast, W_vs_L1 (105) had the fewest DELs, with 58 and 47 DELs up- and downregulated, respectively. In addition, W_vs_L2 had 226 DELs, with 71 and 155 DELs up- and downregulated, respectively (Figure 8A,B). The Venn statistics of the comparative analysis among groups showed that the number of DELs shared between the L1_vs_L2 and W_vs_L2 was 69, while the number of DELs shared with the L1_vs_L2 and W_vs_L1 was 39 and the number of DELs shared between the W_vs_L1 and W_vs_L2 was 28 (Figure 8A). Moreover, comparative analysis of the three groups showed that there were only 12 common DELs (Figure 8A). Further analysis showed that there was no significant difference in the proportion of eight major lipids in the total lipids between the wild plants and OE plants (Appendix A and Figure 8C). However, the proportion of some lipid subclasses among the eight major lipids had changed significantly. Our results showed that most DELs were mainly concentrated in fatty acyls and glycerophospholipids, of which the proportion of two phosphatidyl inositols (PI) and one phosphatidyl choline (PC) increased. (Appendix A).

## 3. Discussion

As a multigene family of basic proteins, nsLTPs play an important role in plant growth and environmental stress. So far, the *nsLTP* gene family has been reported in many plant species, such as poplar (*Populus* L.) [37], sugarcane (*Saccharum* spp.) [23], foxtail millet (*Setaria italica* L.) [20], and Chinese white pear (*Pyrus bretschneideri*) [38]. Studies on the *nsLTP* gene family in chestnut have not been reported yet. In the present study, a total of 60 *CmnsLTPs* were identified in the chestnut genome, and they were divided into nine subfamilies by phylogenetic analysis (Figure 1). Similar gene structural patterns were observed in the same subfamilies, strongly supporting their close evolutionary relationship (Figure 2). However, the genes of the same subfamily showed different tissue-specific expressions, which suggested that they might also play distinct roles in the growth and development of chestnut. As the tissue-specific expression results indicate, most *CmnsLTPs* were expressed primarily at different stages of male flower development, and they were deduced as being involved in pollen development (Figure 4A). Moreover, *nsLTP* genes have been proven to play important roles in abiotic stress. With the goal of screening candidate drought- and osmotic stress-responsive *CmnsLTPs*, an analysis of the expression profiles of 11 *CmnsLTPs* was performed. As shown in Figure 4B, all 11 selected *CmnsLTPs* were responsive to osmotic and drought stress. Additionally, in terms of the promoter elements identified in the *CmnsLTPs*, the promoters of these genes contained many stress-responsive regulatory elements, explaining their response to drought and osmotic stress (Figure 3). These genes could be good candidates for further elucidating the molecular mechanism of the drought and osmotic stress tolerance of Chinese chestnut.

Alternative splicing (AS) is a gene regulatory mechanism that promotes messenger RNA complexity and proteome diversity. These spliceosomes jointly regulate plant growth and development in a synergistic or antagonistic manner. For example, there are three splice variants (*ARF8.1*, *ARF8.2*, and *ARF8.4*) of *ARF8* synergistically involved in the development of male flowers in *Arabidopsis* [39]. Two splicing variants (*RcCPR5-1* and *RcCPR5-2*) of *RcCPR5* were found to have formed homodimers and heterodimers to resist powdery mildew in *Rosa chinensis* [40]. However, the splice variant of the *SND1* transcription factor negatively regulated *SND1* members in *P. trichocarpa* [41]. In this study, *CmnsLTP6.9* was dramatically induced and upregulated by osmotic and drought treatment, indicating that *CmnsLTP6.9* participated in the response to osmotic and drought stress (Figure 4B). Two splicing variants of *CmnsLTP6.9* (*CmnsLTP6.9L* and *CmnsLTP6.9S*) were cloned. Since CmnsLTP6.9S protein lacks the N-terminal signal peptide, it is classified as a non-secretory protein (Figure 5A and Appendix A). The signal peptides are thought to act on the transmembrane transfer and localization of proteins. As has been reported, *CaPGIP2* has differing subcellular localization due to the lack of coding initiation terminal signal peptide in chickpea [42]. Although *CmnsLTP6.9L* and *CmnsLTP6.9S* contain a conserved domain and showed similar subcellular localization and expression patterns, their function during osmotic and drought stress were different (Figure 6). Therefore, we conclude that the role of the signal peptide is indispensable for *CmnsLTPs*.

Under biotic and abiotic stresses, numerous physiological and biochemical changes occur, such as lipid composition in the membrane and proline and antioxidant accumulation. The activities of ROS-scavenging enzymes, such as SOD and POD, play essential roles in stress responses to biotic and abiotic stress. Our results suggest that under osmotic and drought stress, the H_2_O_2_ concentration in the *CmnsLTP6.9L*-OE lines was significantly lower, consistent with higher SOD and POD activities, which indicates that the elevated ROS-scavenging ability in *CmnsLTP6.9L*-OE lines contributed to the enhanced osmotic and drought tolerance (Figure 7). By integrating the above data, we concluded that *CmnsLTP6.9L* regulated osmotic and drought response partially via improving antioxidative activities, which was in line with the reports that *nsLTP* reduced the accumulation of ROS by minimizing the toxic effects of ROS under abiotic stresses [28].

Lipids are key structural components of cell membranes, and play crucial roles in maintaining membrane integrity and cell signaling, mediating plant responses to stress [43]. In this study, the function of *CmnsLTP6.9L* and *CmnsLTP6.9S* was further validated using a lipidomics assay. The KEGG pathway was enriched in arachidonic acid metabolism (W_vs_L1) and sphingolipid metabolism (L1_vs_L2), which play positive roles in plant defenses [44,45]. *nsLTPs* were responsible for phospholipid transfer between fatty acid junctions and membranes in vitro; *CmnsLTP6.9L*-OE significantly changed the fatty acyl and glycerophospholipid metabolism (Appendix A). Glycerophospholipids are important components of plant biofilms and signal transduction, which play important roles in various abiotic stresses [45]. Compared to WT, most differential glycerophospholipids were upregulated in *CmnsLTP6.9L*-OE lines, including two PIs and one PC (Appendix A). The increasing PI and PC content was conducive to maintaining membrane integrity and fluidity [46,47]. PC enhanced homeostasis in peach seedling cell membranes, and increased the plant’s salt stress tolerance by means of phosphatidic acid (PA) [48]. In addition, W_vs_L2 had more differential lipids compared to W_vs_L1, implying that the functions of *CmnsLTP6.9L* and *CmnsLTP6.9S* in plants were different, which could also explain their different roles under osmotic and drought stress (Figure 8A).

## 4. Materials and Methods

### 4.1. Plant Materials

The chestnut variety ‘Jiangshan No. 2′ was grown in our chestnut germplasm orchard in Wuhan (longitude: 114.22346; latitude: 30.34313). All the trees were healthy and under normal management. To detect the expression level of *CmnsLTP* in the chestnuts, male flowers at four developmental stages (flower bud primordium differentiation stage, perianth formation stage, pollen mother cell and tetrad stage, and pollen grain maturation period) and female flowers without pollination at two developmental stages (ovule differentiation stage and female flower maturation period) were collected, quickly frozen in liquid nitrogen, brought back to the laboratory, and stored at −80 °C. At least 3 replicated biological samples were prepared from 6 trees. *A*. *thaliana* (Columbia-0) seeds were germinated on MS medium and transferred to a growth chamber with a 16 h/8 h light/dark cycle at 23 ± 2 °C. *A. thaliana* grown under normal conditions for one month was used for osmotic and drought treatments. The plants were treated with 250 mM mannitol to mimic osmotic stress. For drought treatment, the plants were grown without any watering for two weeks. Samples for the RNA/biochemical/metabolic analysis were collected from 3 replicates and frozen in liquid nitrogen for later use. Young shoots with consistent growth from ‘Jiangshan No. 2′ were selected for osmosis and drought treatments. The 250 mM mannitol was used for osmotic stress treatment, and leaves with uniform growth were collected at 0, 1, 3, 6, and 24 h after treatment. In addition, the growth of chestnut shoots under anhydrous conditions simulated drought stress, and leaves with uniform growth were collected at 0, 1, 3, 6, and 9 h after treatment. All examples were quickly frozen in liquid nitrogen and stored at −80 °C.

### 4.2. Phylogenetic Analysis of the nsLTP Genes in Chestnut

The whole chestnut protein file was downloaded from the NCBI database (https://www.ncbi.nlm.nih.gov/datasets/genome/GCA_000763605.2/, accessed on 25 January 2022). The reference genome databases of *Arabidopsis thaliana* and *B. napus* are TAIR (https://www.arabidopsis.org/, accessed on 25 January 2022) and BRAD (http://brassicadb.cn/#/Download/, accessed on 25 January 2022), respectively. The hidden Markov model (HMM) profiles PF14368 and PF00234 were used as hmmsearch queries (*p* < 0.001; http://hmmer.org, accessed on 28 January 2022). For hmmsearch, the default parameters were adopted, and the cutoff E-value was <1 × 10^−5^. To avoid the possible loss of the *nsLTP* gene due to an incomplete extracellular matrix (ECM) domain, a local BLASTP was performed with a cutoff E-value of <1 × 10^−5^, using the published *A. thaliana* and *Brassica napus nsLTP* amino acid sequences as queries. After removal of the repeated sequences, all assumed nsLTP proteins were submitted to SignalP 5.0 (http://www.cbs.dtu.dk/services/SignalP/, accessed on 20 May 2023) to confirm the presence of signal peptides. All candidate nsLTP proteins were then submitted to Pfam 32.0 (http://pfam.xfam.org, accessed on 20 May 2023) to confirm the LTP domains and to Batch Web CD-Search Tool (https://www.ncbi.nlm.nih.gov/Structure/bwrpsb/bwrpsb.cgi, accessed on 20 May 2023) to confirm the AAI_LTSS domains. Finally, all candidate *CmnsLTPs* were examined manually to eliminate proteins lacking the essential ECM sequences. The conserved sequence of the ECM was edited manually by MEGA11.0. Then, the phylogenetic relationships of the *nsLTPs* were constructed using the neighbor-joining method by MEGA 11.0 software with the following parameters: Poisson model, pairwise deletion, and 1000 bootstrap replications. The phylogenetic trees were visualized using the iTOL V6 (https://itol.embl.de/, accessed on 20 May 2023). All the gene accession numbers and protein sequences are listed in Appendix A.

### 4.3. Conserved Motifs, Gene Structure, and Promoter Analysis

The conserved motifs and gene structure of nsLTP proteins were analyzed using MEME Suite 5.5.2 online program (Multiple Em for Motif Elicitation 5.5.2, http://alternate.meme-suite.org/tools/meme, accessed on 25 May 2023), and the gene structure and signal peptide were analyzed via TBtools 2.021 and SignalP-5.0 (https://services.healthtech.dtu.dk/service.php?SignalP-5.0, accessed on 25 May 2023). The promoter sequences comprising 2000 bp of the upstream regions of coding initiation terminal of *CmnsLTPs* were extracted. Cis-acting elements were predicted via PlantCARE (http://bioinformatics.psb.ugent.be/webtools/plantcare/html/, accessed on 26 May 2023), and the responsive regulatory elements were analyzed via TBtools.

### 4.4. RNA Extraction and qRT-PCR Assay

Total RNA was extracted from the collected samples using the RNA prep Pure Plant Kit (Tiangen, Beijing, China) and cDNA was prepared using Monad MonScriptTMRTIII Super Mix with dsDNase (Two-Step) (Monad, Suzhou, China). The qRT-PCR analysis was performed in Applied Biosystems PCR-7500 (ABI, Carlsbad, CA, USA) by Monad MonAmpTM SYBR^®^ Green qPCR Mix (Monad, Suzhou, China). The primers used for the qRT-PCR are shown in Appendix A. Three replicates were conducted for the assay.

### 4.5. Subcellular Localization of CmnsLTP6.9S and CmnsLTP6.9L

To construct the 35S: *CmnsLTP6.9*-GFP expression plasmid, the CDS of *CmnsLTP6.9S* and *CmnsLTP6.9L* was inserted into the binary vector pMD18-T. The recombinant plasmid pMD18-T-*CmnsLTP6.9* and empty vector pCAMBIA2300-GFP were cut with XbaI and Sac I enzymes and linked with T4 ligase, and the plant expression vector pCAMBIA2300-CmnsLTP6.9-GFP was obtained. Both 35S:GFP and the recombinant vectors were transferred into EHA105. Positive clones were selected to shake the bacteria and infect the onion epidermal cells. After 3 days of dark culture on MS medium, green fluorescence was observed using a fluorescence microscope INTENSILIGHT C-HGFI (Nikon, Tokyo, Japan). All of the primers are shown in Appendix A.

### 4.6. Genetic Transformation and Fluorescence Analysis of A. thaliana

To validate the potential function of *CmnsLTP6.9S* and *CmnsLTP6.9L*, the positive binary vectors used for the subcellular localized assay were applied for genetic transformation in *A. thaliana* with the floral dip method. Third-generation (T3) seedlings of the overexpressing (OE) lines and wild-type (WT) *Arabidopsis* plants were used for subsequent functional verification. Detection of green fluorescence in the wild-type and transgenic *Arabidopsis thaliana* was conducted using a dual fluorescent protein observation lamp LUYOR-3415RG (LUYOR, Shanghai, China) under dark conditions 15 days after germination on the culture medium.

### 4.7. Osmotic and Drought Treatment

One-month-old *A. thaliana* of the WT and OE lines were irrigated with or without 250 mM mannitol or water solution for 2 weeks. The relative water content and relative electrolyte leakage were analyzed after osmotic and drought treatment. The contents of proline and malondialdehyde (MDA) were tested with the proline assay kit and the MDA assay kit (Jiancheng Bioengineering Institute, Nanjing, China). Three replicates were conducted for the assay.

### 4.8. Oxidative Stress Analysis

Staining was performed using DAB and NBT to detect the accumulation of H_2_O_2_ and O_2_^−^. A hydrogen peroxide test (Jiancheng Bioengineering Institute, Nanjing, China) was used to detect the H_2_O_2_ concentrations. The activities of the antioxidant enzymes SOD and POD were measured with the kits from Nanjing Jiancheng Institute. Each assay was repeated three times.

### 4.9. Lipidomics Analysis

About 50 mg of fresh leaves of *Arabidopsis* (*CmnsLTP6.9S*-OE, *CmnsLTP6.9L*-OE, and WT lines) were pre-treated with 480 μL of extract solution (methyl tert-butyl ether:methanol = 5:1). After sonication, centrifugation, and drying, 200 μL of solution (dichloromethanes:methanol = 1:1) was added for reconstitution. The mixture was then centrifuged for 15 min at 12,000 rpm and the organic layer (~200 μL) was transferred to a clean glass sample bottle for UHPLC-QTOF-MS non-target lipidomics assay. An ultra-performance liquid chromatographer (Acquity I-Class PLUS) (Waters, Milford, CT, USA) and a high-resolution mass spectrometer (Xevo G2-XS QTof) (Waters, Milford, CT, USA) were used for the metabolomic analysis. The chromatographic column was an Acquity UPLC CSH C18 (1.7 μm, 2.1 mm × 100 mm) (Waters, Milford, CT, USA). Positive ion mode: mobile phase A: 60% acetonitrile aqueous solution, 10 mM ammonium acetate, 0.1% formic acid; mobile phase B: 90% isopropanol acetonitrile solution, 10 mM ammonium acetate, 0.1% formic acid anion mode: mobile phase A: 60% acetonitrile solution, 10 mM ammonium acetate, 0.1% formic acid; mobile phase B: 90% isopropyl alcohol acetonitrile solution, 10 mM ammonium acetate, 0.1% formic acid. The injection volume was 5 μL and the column temperature was 55 °C. The low collision energy was 2 V, the high collision energy range was 10~40 V, and the scanning frequency was 0.2 s for one mass spectrum. The ESI ion source parameters were as follows: capillary voltage: 2000 V (positive ion mode) or −1500 V (negative ion mode); cone voltage: 30 V; ion source temperature: 120 °C; desolvation gas temperature: 550 °C; backflush gas flow rate: 50 L/h; desolvation gas flow rate: 900 L/h. MassLynx V4.2 and Progenesis QI 3.0 were used to collect and process the data. The specific lipid analysis referred to the method reported previously [49]. All of the lipidomics are shown in Appendix A.

### 4.10. Statistical Analysis

SPSS 20 software (IBM, Armonk, NY, USA) was used to perform the analysis of One-way ANOVA (Duncan), and statistically significant differences are indicated by different lowercase letters (*p* < 0.05).

## 5. Conclusions

In total, 60 *CmnsLTPs* belonging to nine subfamilies were identified by genome-wide analysis of chestnut; their expression pattern revealed that the *CmnsLTPs* were tissue specific, and 11 *CmnsLTPs* were involved in growth and tolerance to drought and osmotic stress. Specifically, we demonstrated that one splicing variant of *CmnsLTP6.9* enhanced drought and osmotic stress tolerance by regulating ROS clearance and lipid metabolism (Figure 9). The other variant of *CmnsLTP6.9* lost similar functions due to the lack of a signal peptide. Our work provides new insights into *nsLTPs* in chestnut, and the genes screened in the study would be valuable candidates for osmotic and drought tolerant breeding in the future. However, many concerns need to be further addressed in the future, such as what the lipid substrate transferred by *CmnsLTP6.9L* and *CmnsLTP6.9S* is and whether *CmnsLTP6.9L* and *CmnsLTP6.9S* are involved in pollen maturation and other stresses.

## Figures and Tables

**Figure 1 plants-12-03916-f001:**
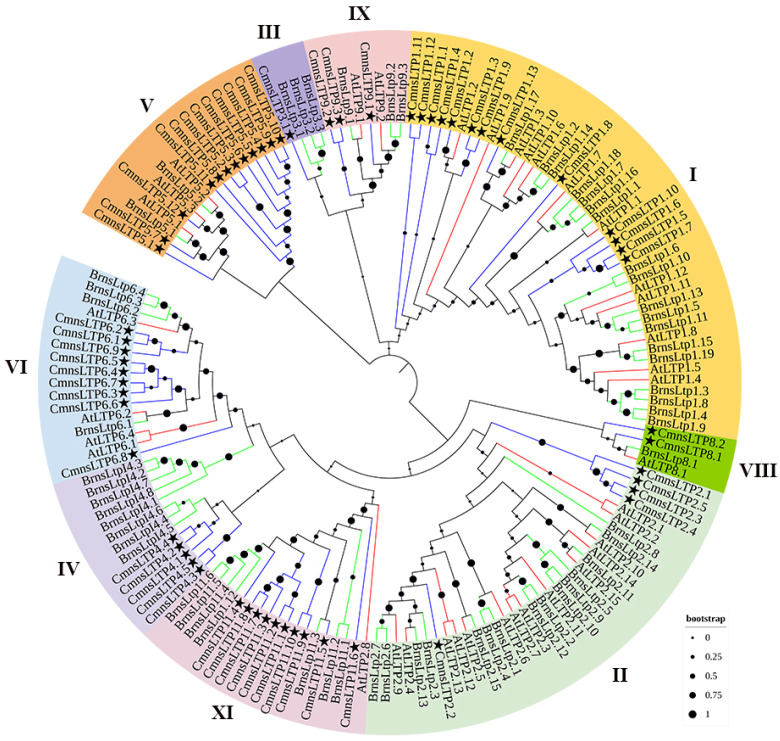
Phylogenetic tree of *nsLTP* proteins from *C. mollissima*, *A. thaliana*, and *B. napus*. Nine *nsLTP* protein types are marked using different colors. Stars represent the genes of *C. mollissima*. Blue lines represent the genes of *C. mollissima*. Green lines represent the genes of *B. napus*. Red lines represent the genes of *A. thaliana*. Black dots represent the clades’ support values in the phylogenetic trees.

**Figure 2 plants-12-03916-f002:**
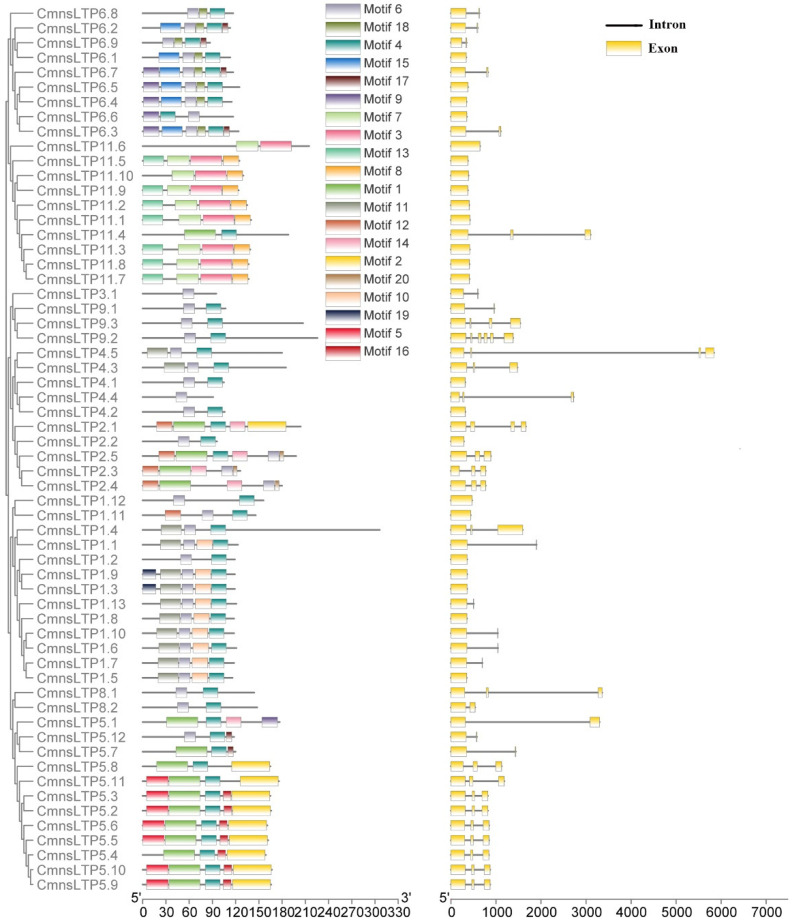
Gene structure and motif compositions of *CmnsLTPs*. **Left**: conserved motif composition of *CmnsLTPs*. The different colored boxes represent different motifs. The scale bar at the bottom represents 30 aa. **Right**: intron–exon structure of *CmnsLTPs*. Yellow boxes represent exons; gray lines represent introns. The scale bar at the bottom represents 1000 bp.

**Figure 3 plants-12-03916-f003:**
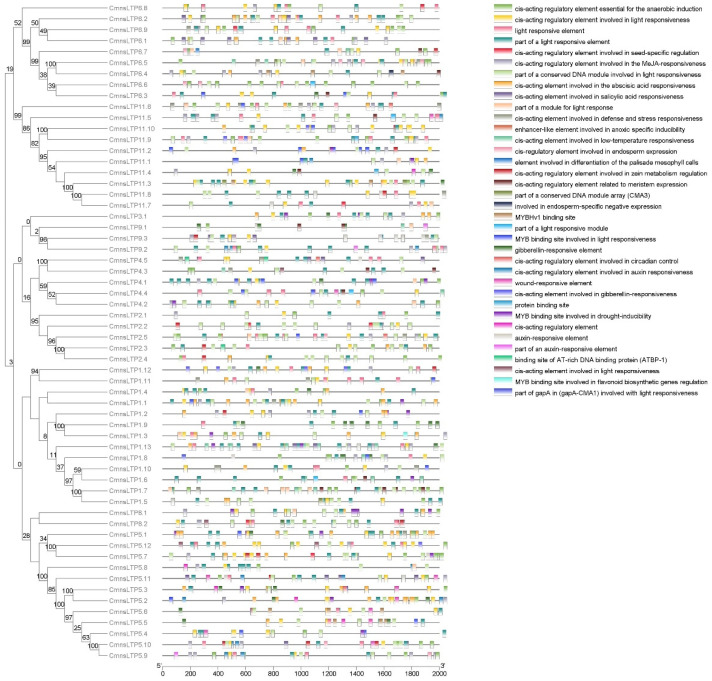
Responsive cis-acting elements predicted in the *CmnsLTPs* promoters. Different colors represent different responsive elements.

**Figure 4 plants-12-03916-f004:**
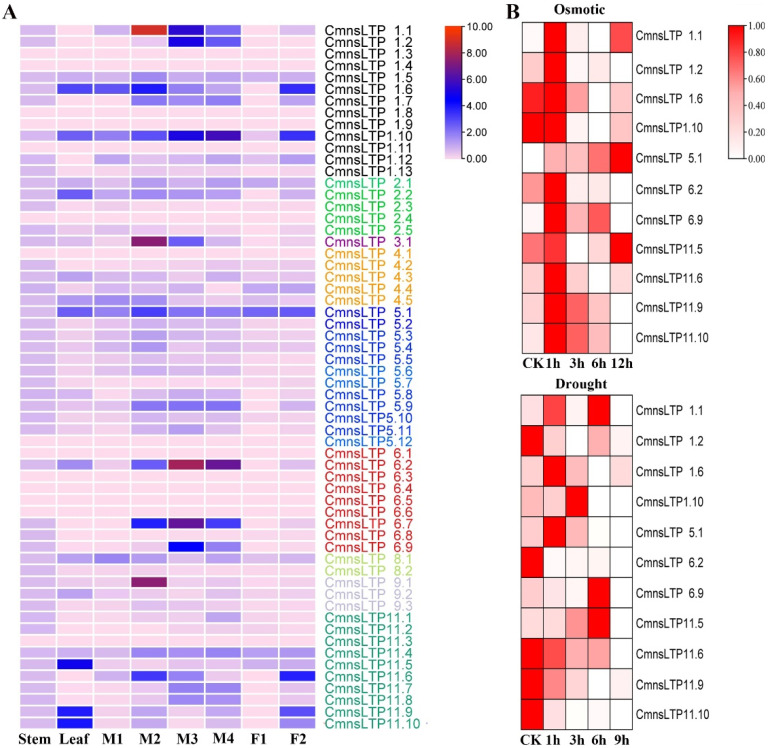
Expression profiling of *CmnsLTPs.* (**A**) Expression analysis of *CmnsLTPs* in various tissues of chestnut. The visualization of the result was achieved using TBtools. Different colored *CmnsLTPs* represent different expression patterns. The color scale represents the relative signal intensity. M1, M2, M3, and M4 represent male chestnut flowers at four developmental stages. M1, flower bud primordium differentiation stage; M2, perianth formation stage; M3, pollen mother cell and tetrad stage; M4, pollen grain maturation period. F1 represents the female chestnut flower at the ovule differentiation stage; F2, maturation period. (**B**) Heat map representation of *CmnsLTPs* under osmotic (0 h, 1 h, 3 h, 6 h, and 24 h) and drought (0 h, 1 h, 3 h, 6 h, and 9 h) stress from the qRT-PCR experiment. The color bar represents the relative signal intensity value.

**Figure 5 plants-12-03916-f005:**
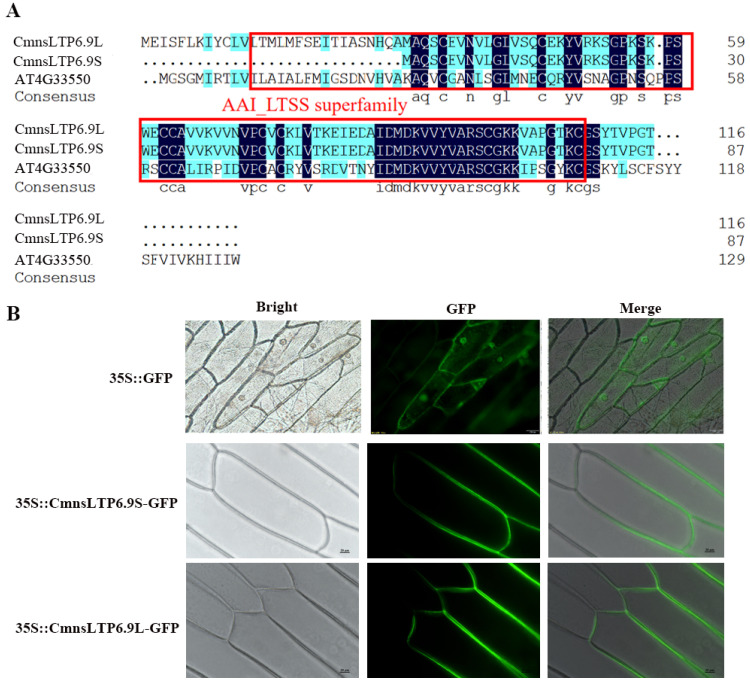
Protein sequence alignment and subcellular localization analysis. (**A**) Protein sequence alignment of *CmnsLTP6.9S* and *CmnsLTP6.9L*. The sequences in the red boxes are conserved domains. Black highlights represent the highly conserved protein sequence and blue highlights represent the different protein sequence between *C. mollissima* and *A. thaliana*. (**B**) Transient expression of *CmnsLTP6.9S*-GFP and *CmnsLTP6.9L*-GFP in onion epidermal cells. Bars = 20 μm.

**Figure 6 plants-12-03916-f006:**
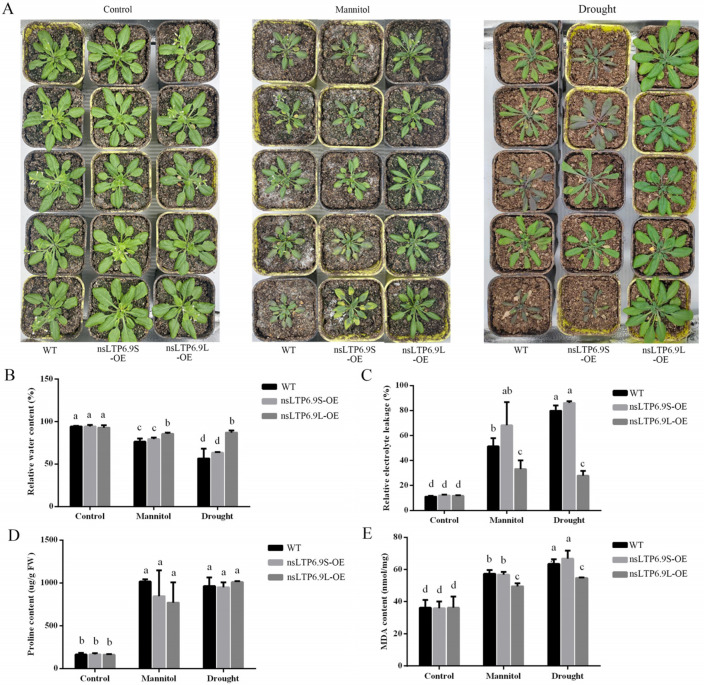
Overexpression of *CmnsLTP6.9L* in transgenic plants enhanced for osmotic and drought tolerance. (**A**) Phenotype of 1-month-old WT and OE plants grown in soil irrigated with water, 250 mM mannitol solution, and without water for 2 weeks. (**B**) Relative water content. (**C**) Relative electrolyte leakage. (**D**) Proline content. (**E**) MDA content. The data are the mean values of three biological repeats. The error bars indicate the SE. One-way ANOVA (Duncan) was performed, and statistically significant differences are indicated by different lowercase letters (*p* < 0.05).

**Figure 7 plants-12-03916-f007:**
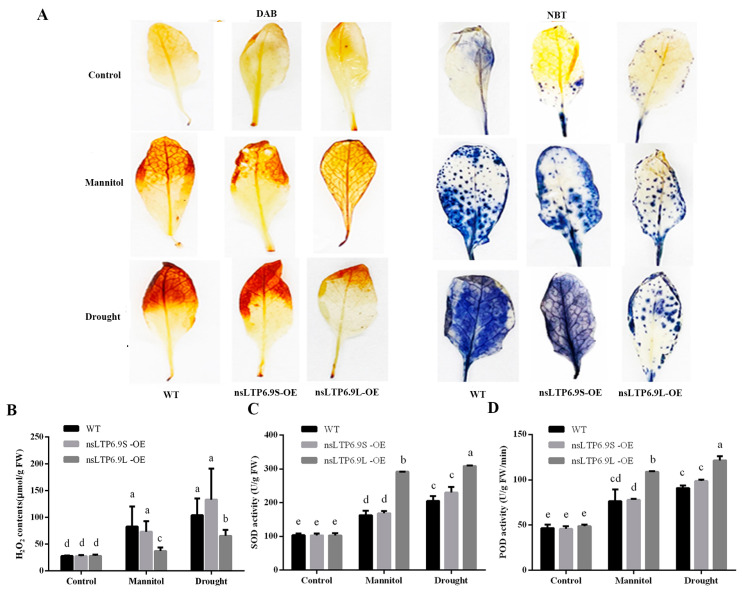
The overexpression of *CmnsLTP6.9L* in transgenic plants decreased the ROS content and oxidative damage under osmotic and drought treatment. (**A**) Photographs of DAB and NBT staining in WT and OE plants under osmotic and drought treatment. (**B**) H_2_O_2_ concentrations measured by the kit. (**C**,**D**) SOD and POD activities are measured by the corresponding kits, respectively. The error bars indicate the SE. One-way ANOVA (Duncan) was performed, and statistically significant differences are indicated by different lowercase letters (*p* < 0.05).

**Figure 8 plants-12-03916-f008:**
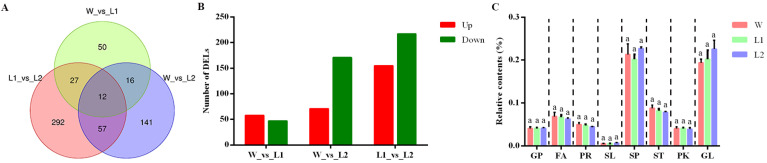
Statistics on the numbers of DELs. (**A**) Venn diagram statistics of the comparisons among the different groups. W, WT; L1, *CmnsLTP6.9L*; L2, *CmnsLTP6.9S*. (**B**) Statistical analysis of DELs in the different groups in the different plants. (**C**) Relative contents of different lipids in the plants. GP, glycerophospholipids; FA, fatty acyls; PR, prenol lipids; SL, saccharolipids; SP, sphingolipids; ST, sterol lipids; PK, polyketides; GL, glycerolipids. The dotted lines represent inter-group intervals and intra-group comparisons. The error bars indicate the SE. One-way ANOVA (Duncan) was performed, and statistically significant differences are indicated by different lowercase letters (*p* < 0.05).

**Figure 9 plants-12-03916-f009:**
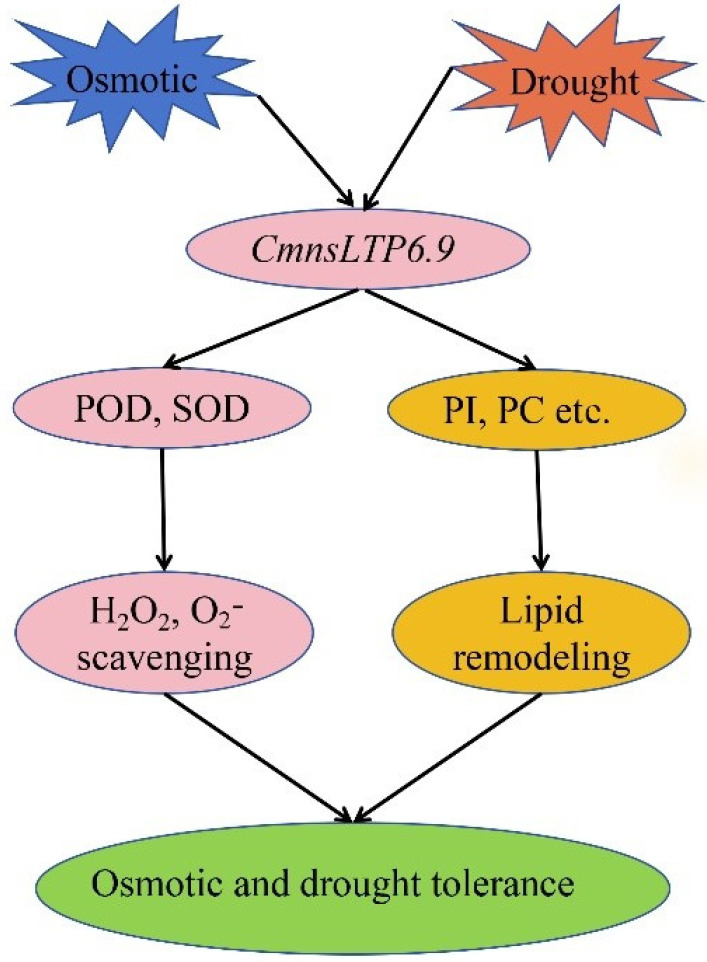
A potential model of *CmnsLTP6.9* regulating plant responses to osmotic and drought stress. Osmotic and drought stress induced the expression of *CmnsLTP6.9*. After osmotic and drought stress, *CmnsLTP6.9*-OE increased the activities of POD and SOD in *Arabidopsis thaliana*, thus scavenging H_2_O_2_ and O_2_^−^. However, *CmnsLTP6.9*-OE remodeled the lipid content of *A. thaliana*, thus increasing the content of PI and PC and promoting tolerance to drought and osmotic stress.

## Data Availability

Data are contained within the article and Appendix A.

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
