# Peer review of "A Novel Non-Specific Lipid Transfer Protein Gene, CmnsLTP6.9, Enhanced Osmotic and Drought Tolerance by Regulating ROS Scavenging and Remodeling Lipid Profiles in Chinese Chestnut (Castanea mollissima Blume)"

_plants, 2023, doi:10.3390/plants12223916_

Round 1
Reviewer 1 Report
Comments and Suggestions for Authors
The research topic is new and interesting, but the authors’ choice of gene remains unclear. Analysis of research results should be improved. Here are the suggestions for authors:
1. You are often confusing genes and proteins through the text
"Results showed that CmnsLTP6.9 localized
in the extracellular matrix, had two splicing variants"
Is gene localized in extracellular matrix??
"Compared
with CmnsLTP6.9L, CmnsLTP6.9S had an 87-bp deletion in the N-terminal"
Does a gene have N-terminal?
2. "CmnsLTP6.9L en-
Ehanced drought and osmotic stress tolerance through ROS scavenging and lipid metabolism."
Were ROS scavenging and lipid metabolism increased or decreased, what's the mechanism? It is not clear from the abstract. I also couldn't find the mechanism of action of these LTPs 6.9 in the whole text. How exactly they affect drought and osmotic stress tolerance? Please add a scheme as a Figure.
It is also not clear why did you choose and clone LTP6.9 exactly? Why not another one? According to your data, this LTP is highly expressed during pollen maturation. But you collected only leaves to study the response to stress ("leaves with uniform growth were collected at 0, 1, 3, 6 and 9h after treatment.")
3. "drought and osmotic stress are two common abiotic
stresses that limit the growth and development of chestnuts and affect fruit quality and
yield."
This statement is not supported by any reference. How exactly do these stress factors affect chestnuts? How much yield is lost? What is the difference in fruit quality?
4. "the published A. thaliana and Brassica rapa nsLTP amino acid sequences"
give a link to these sequences
Why didn't you use sequences of a tree as a reference?
5.
"To construct the 35S::CmnsLTP6.9-GFP expression plasmid, the CDS of CmnsLTP6.9S
and CmnsLTP6.9L was inserted into the binary vector pCAMBIA2300-GFP."
please provide the map of the resulting vector
Did you fuse CmnsLTP6.9 to GFP? What linker did you use? What restriction enzyme did you use for cloning?
6.
"4.9 Lipidomics analysis"
The description of analysis is not full. For example, you did not specify the solvent inside a column. You should also give a reference to the method
7.
"Excluding down-regulated genes, CmnsLTP1.1, CmnsLTP5.1, CmnsLTP6.9 and
CmnsLTP11.5 positively responded to osmosis and drought treatment, in which
CmnsLTP1.1, CmnsLTP5.1 and CmnsLTP11.5 was highly expressed in the pre-and post-
stage"
According to Fig 5, genes 11.9 and 11.10 were upregulated in the same way as 6.9. Why didn't you choose them?
8.
"Transgenic lines were verified by RT-PCR and green fluo-
rescence, of which five lines with the highest expression levels were used for subsequent
functional analysis"
You do not provide information, how did you analyze fluorescence in A. thaliana.
According to Fig. S2, in lines L4 and S1 transgene expression level is at least 3 times higher than in other lines. Were there any differences between these two lines and other lines? You should study this because the difference in expression level is really big.
Comments on the Quality of English Language
Language needs editing
"adoptability to environmental stresses especially drought."
comma missing
"lipid transfer proteins (LTPs) genes responded to abiotic stress in chestnut."
responding
"CmnsLTP6.9L mainly altered and up-regulated many
fatty acyls and glycerophospholipids lipids"
Are not altering and up-regulation the same thing in this context? Did these genes increase the activity or the content pf these molecules?
"In addition
to the kernels for food processing, chestnut shells, flowers and leave were widely used for
medical, biomass and catalyst materials"
leaves? medical what? biomass what?
"chestnut trees are resistant to barrenness"
Do you mean that they are evergreen?
"the published A. thaliana and Brassica rapa nsLTP amino acid sequences"
latin should be in italic. Check through the text
"vectors used for subcellular localized assay"
localization assay?
"After sonication, centrifugation and drying, add 200 μL solution (dichloromethane: meth-
anol=1:1) for reconstitution."
solution WAS ADDED?
On Fig S2 "relative expression lever"
level
there are also no units of measurement (is it %?)
Author Response
Thank you for your letter and the comments concerning our manuscript entitled “plants-2640715”. Those comments are all valuable and very helpful for revising and improving our paper, as well as the important guiding significance to our researches. We have studied the comments carefully and have made corrections which we hope meet with approval. The main corrections can be immediately recognized with red markers and revision mode in the paper and the responds to the reviewer’s comments are in the attached file.

Reviewer 2 Report
Comments and Suggestions for Authors
This is an interesting report with some factual mistakes, and with very bad English (see below). It requires major revision before being resubmitted.
line 30 'woody crops' is NOT a name for food crops that grow on trees. It is a name for crops that are grown for wood.
Figure 1. unclear, hard to read
Figure 4. How are there pollen grains in female flowers? They should be only in male flowers until fertilization, and then many genes change in expression.
Figure 9-- there are really no differences in lipid composition between the two engineered lines and the wild type. Despite that, the authors hint in their conclusion (lines 323-332) that there are differences in glycolipid content, but the data do not support this notion. The engineered/trasnformed seedlings do survive drought , do have greater relative water content and reduced electrolyte leakage and MDA content, but the reason for this is not to be found in lipid content or structure, at least not from the data presented in the manuscript. The rewritten manuscript should highlight only facts that are supported by the data.
Comments on the Quality of English LanguageI read the whole manuscript, but made comments only until I got tired. The English in the manuscript MUST be edited by a GOOD editor or even by Chat GPT or Grammarly or similar programs. Don't you even use spellcheck? Below are some comments before I decided that the whole mss needs to be rewritten.
line 14 tested--> tasty
15 adoptability--> adaptability
17 what does ns mean in nsLTP?
64 evidence
78 benefited significantly? what does that mean in this sentence?
269 poplar, foxtail millet-- mind your Capital Letters
There are many many many other cases of poor spelling, wrong words, bad grammar. Please fix.
Author Response

(The authors gave the same response as above.)

Reviewer 3 Report
Comments and Suggestions for Authors
The article by Yu-Xiong Xiao, Cui Xiao, Xiu-Juan He, Xin Yang, Zhu Tong, Ze-Qiong Wang, Zhong-Hai Sun and Wen-Ming Qiu is devoted to studying the role of two splicing variants of the gene encoding nonspecific lipid transfer proteins (nsLTP), under the influence of osmotic and drought stress in Castanea mollissima Blume. It has been shown that overexpression of the variant without deletion CmnsLTP6.9L increases plant resistance to abiotic factors due to the activation of enzymes that reduce the content of reactive oxygen species and the intensification of fatty acid metabolism. The article provides its own original data. The title, abstract and keywords objectively reflect the content of the article.
Comments on the article:
1) Lines 76-91. A paragraph goes to the abstract of the article, at the same time there is no null hypothesis.
2) Lines 102-105. Italics: nsLTP, C. mollissima.
3) Line 137. Italics: CmnsLTP.
4) Line 166. Replace “pre-and” with “pre- and”.
5) Line 167. The conclusion “only CmnsLTP6.9 was highly expressed in the early and middle stages” follows unconvincingly from Fig. 5. So, for example, in my opinion, CmnsLTP1.1 or CmnsLTP11.5 have the same characteristics.
6) Line 172. Add the designations CK, 1h, 3h, 6h, 24h to the captions of Figure 5.
7) Lines 194, 245, 401. Italics: Arabidopsis.
8) Lines 219, 226, 227, 239. Replace “H2O2” with “H2O2”.
9) Line 237. Italic: CmnsLTP6.9L.
10) Line 244. Replace “Lipidomic” with “lipidomic”.
11) Line 246. Replace “p-value less than 0.05” with “p<0.05”.
12) Lines 267, 268, 270, 278, 358, 361, 374. Italics: nsLTP.
13) Lines 291, 292. Italics: RcCPR5-1, RcCPR5-2, RcCPR5.
14) Line 293. Replace “Splice” with “splice”.
15) Line 323. Replace “and Glycerophospholipids” with “and glycerophospholipids”.
16) Line 325. Replace “Phosphatidyl inositols” with “phosphatidyl inositols”.
17) Line 326. Replace “Phosphatidyl choline” with “phosphatidyl choline”.
18) Line 329. Replace “Phosphatidic Acid (PA)” with “phosphatidic acid”.
19) Line 360. Italics: A. thaliana, Brassica rapa, sLTP.
20) Line 376. Replace “Peptide” with “peptide”.
21) Line 400. Italics: A. thaliana.
22) The methodology does not have a section on statistical methods of data analysis.
Author Response

(The authors gave the same response as above.)

Reviewer 4 Report
Comments and Suggestions for Authors
Manuscript ID: plants-2640715
Here, a chestnutns LTP, named CmnsLTP6.9 was identified and analyzed。The overall feeling is that the paper is written more clearly. But there are also some problems in the paper, and I hope the author can revise it.
In the result section, the Author always added some unnecessary statements in front of the order to do something (please see the yellow marks), which can be deleted; Line 94, 107-108, 128-130, 140-142, 175-176, 193-194, 218-219,244;
Pay attention to superscripts and subscripts, such as H2O2, O2-; Line 219-239.
It is suggested that Figure 4 and Figure 5 be combined into one large Figure 4. They were arranged left and right, and delete one the relative signal intensity value. It can save publishing spaceï¼›
Line142. “suggested” change as “indicated”ï¼›
Line 269, Add the latin name after the scientific name;
Figure 8 is not clear, it is recommended to modify Figure 8, put Figure 8a on top, enlarge slightly, Figure b, c, d put below;
Line 341, 343, 352, temperature shows an error about it oC;
The equal sign "=" leave a space on the left and right sides.
Manuscript ID: plants-2640715
Here, a chestnutns LTP, named CmnsLTP6.9 was identified and analyzed。The overall feeling is that the paper is written more clearly. But there are also some problems in the paper, and I hope the author can revise it.
In the result section, the Author always added some unnecessary statements in front of the order to do something (please see the yellow marks), which can be deleted; Line 94, 107-108, 128-130, 140-142, 175-176, 193-194, 218-219,244;
Pay attention to superscripts and subscripts, such as H2O2, O2-; Line 219-239.
It is suggested that Figure 4 and Figure 5 be combined into one large Figure 4. They were arranged left and right, and delete one the relative signal intensity value. It can save publishing spaceï¼›
Line142. “suggested” change as “indicated”ï¼›
Line 269, Add the latin name after the scientific name;
Figure 8 is not clear, it is recommended to modify Figure 8, put Figure 8a on top, enlarge slightly, Figure b, c, d put below;

No
Author Response

(The authors gave the same response as above.)

Round 2
Reviewer 1 Report
Comments and Suggestions for Authors
Dear authors!
The manuscript became better, but one of the most important questions still doesn't have an answer
“According to Fig. S2, in lines L4 and S1 transgene expression level is at least 3 times higher than in other lines. This may be related to the insertion of genes in plants, and similar situations have been observed in many studies. This phenomenon has research significance, but it is not related to the content of this study, and this expression difference will not affect the study of CmnsLTP6.9 function”
If transgene really affects stress tolerance, there should be a difference in reaction to stress between plants with higher and lower expression of transgene. Were there any differences in stress resistance between the transgenic lines?
Comments on the Quality of English Language
The formatting of changes is confusing. Corrected errors are not crossed out in the text; corrections are written next to them. For example,
“lipids, which implemented implied that”
“genes that responded respond to abiotic stress”
Please check through the text that corrections are applied.
“After sonication, centrifugation and drying, add 200 μL solution (dichloromethane: methanol = 1:1) for reconstitution. ”
solution was added? Not “add”
Reviewer 2 Report
Comments and Suggestions for Authors
The authors had somebody do a mediocre job of English editing, then they did not bother to produce a finished manuscript. It is impossible to read with three colors of letters-- are we supposed to read the red or the green? If they had crossed-out words it would have been MUCH easier to understand.
line 38 Nuts are NOT grains. Please get a competent editor or at least a good dictionary. Better yet, rewrite the sentence without the part about complementary foods, which is irrelevant to the topic of the paper.
Line 170 still refers to female flowers with pollen. This is biologically impossible.
line 381 Authors say that they used 250mM mannitol and a water deficit (I think they mean to induce drought or osmotic stress) In line385 they refer to "approximately 250 mM mannitol" for osmotic stress treatment. Did they measure the mannitol or not? It is not hard to make a 250 mM solution precisely.
There are now two Figures 4, making it difficult to tell which one the authors mean in their discussion. In any event, the authors do not specifically refer to any of their figures in the Discussion, except one reference to Figure 5. This makes it hard to refer to the data that they are trying to explain.
The captions in figures 6 and 7 state that means were separated by Duncan's LSD. Figure 8 has SD or SE bars, but no indication of the statistic they represent. Line 489 says that Fisher's LSD or Student's t-test was used, although the t-test is not applicable in any of the data sets presented.
The manuscript is very hard to read in its current format. The English level, as best as the mss. can be read, is not significantly improved. The graphs are indeed easier to read, but no easier to understand. The statistics are questionable and discussion does not relate sufficiently to specific data.
The manuscript cannot be accepted in its current form.
Comments on the Quality of English LanguageThe manuscript is very hard to read in its current format with the previous and current versions presented at once in red and green. The English level, as best as the mss. can be read, is not significantly improved.
Reviewer 4 Report
Comments and Suggestions for Authors
Leave a space between number and unit.

No comment.
Round 3
Reviewer 2 Report
Comments and Suggestions for Authors
The manuscript is much improved.
In line 355 you wrote "And for drought treatment, the plants grow with a water deficit for two weeks."
Does this mean that plants were grown without any watering for two weeks, or with a lower amount of water (how much less, by percentage?)?
Comments on the Quality of English LanguageThe English is much improved. The only correction to be made is line 355-- you wrote "And for drought treatment, the plants grow with a water deficit for two weeks."
Does this mean that plants were grown without any watering for two weeks, or with a lower amount of water (how much less, by percentage?)?
"Drought was induced by growing the plants with a water deficit of X% for two weeks"
